# Comparative Analysis of Peri-Implant Bone Loss in Extra-Short, Short, and Conventional Implants. A 3-Year Retrospective Study

**DOI:** 10.3390/ijerph17249278

**Published:** 2020-12-11

**Authors:** Daycelí Estévez-Pérez, Naia Bustamante-Hernández, Carlos Labaig-Rueda, María Fernanda Solá-Ruíz, José Amengual-Lorenzo, Fernando García-Sala Bonmatí, Álvaro Zubizarreta-Macho, Rubén Agustín-Panadero

**Affiliations:** 1Department of Stomatology, Faculty of Medicine and Dentistry, University of Valencia, 46010 Valencia, Spain; dayceli.estevez@uv.es (D.E.-P.); naiabustamante@gmail.com (N.B.-H.); m.fernanda.sola@uv.es (M.F.S.-R.); jose.amengual@uv.es (J.A.-L.); fernando.garcia-sala@uv.es (F.G.-S.B.); ruben.agustin@uv.es (R.A.-P.); 2Department of Implant Surgery, Faculty of Health Sciences, Alfonso X el Sabio University, 28691 Madrid, Spain; amacho@uax.es

**Keywords:** short dental implants, marginal bone loss, tissue-level, peri-implantitis, implant-supported prostheses

## Abstract

Objective: To evaluate the influence of implant length on marginal bone loss, comparing implants of 4 mm, 6 mm, and >8 mm, supporting two splinted crowns after 36-month functional loading. Materials and Methods: this retrospective clinical trial evaluated the peri-implant behavior of splinted crowns (two per case) on pairs of implants of the same length placed in the posterior maxilla (molar area). Implants were divided into three groups according to length (Group 1: extra-short 4 mm; Group 2: short 6 mm; Group 3: conventional length >8 mm). Marginal bone loss was analyzed using standardized periapical radiographs at the time of loading and 36 months later. Results: 24 patients (19 women and 5 men) were divided into three groups, eight rehabilitations per group, in the position of the maxillary first and second molars. The 48 Straumann^®^ Standard Plus (Regular Neck (RN)/Wide Neck (WN)) implants were examined after 36 months of functional loading. Statistical analysis found no significant differences in bone loss between the three groups (*p* = 0.421). No implant suffered biological complications or implant loss. Long implants were associated with less radiographic bone loss. Conclusions: extra-short (4 mm); short (6 mm); and conventional length (>8 mm) implants in the posterior maxilla present similar peri-implant bone loss and 100% survival rates in rehabilitation, by means of two splinted crowns after 36 months of functional loading. Implants placed in posterior positions present better bone loss results than implants placed in anterior positions, regardless of the interproximal area where bone loss is measured. Conventional length (>8 mm) implants show better behavior in terms of distal bone loss than short (6 mm) and extra-short (4 mm) implants.

## 1. Introduction

Physiological resorption of bone volume begins to develop as soon as teeth are lost from the mandible or the maxilla. This is due to the fact that formation and preservation of the alveolar processes depend on the presence of the teeth. In particular, the maxilla undergoes centripetal resorption from vestibular to palatine, while resorption in the mandible is centrifugal from lingual to vestibular [1,2]. Nevertheless, whether in the mandible or the maxilla, more or less resorption will occur, depending on the number of teeth lost and the time passed since tooth loss [3].

The classic approach to restoring the masticatory function, speech, and esthetics of the stomatognathic system has been through fixed or removable partial dentures. While this type of treatment remains common and completely valid, the literature reports that implant-supported treatments are overtaking traditional forms of rehabilitation [4,5,6]. Advances in osseointegration, diverse implant designs, and the growing potential to individualize treatment to meet the specific needs of the patients have made implant-based treatment the preferred rehabilitation option [7].

For many years now, the use of implant-supported prostheses has been the treatment of choice in terms of restoring masticatory function and esthetics in partially or totally edentulous patients, especially in distal edentulous areas. However, the insertion of standard-length implants in maxillary posterior regions presents a particular challenge because of the presence of certain anatomical limitations, for example, insufficient bone volume, poor bone quality, limited visibility, reduced inter-arch space, the proximity of the inferior dental nerve or mentonian nerve, among others. In addition, these difficult anatomical conditions appear to worsen when patients have previously worn a removable denture. This may be explained by the bone remodeling that takes place when the bone is not subject to occlusal forces, unlike implant placement, in which osseointegration plays a fundamental role in reducing bone resorption [7,8,9,10,11,12]. With the aim of overcoming vertical, centrifugal, and centripetal resorption in the maxilla, numerous studies have described techniques employed to improve anatomical conditions around the implant bed, in terms of both bone quantity and quality [11,13]. The most frequently described procedures are: guided bone regeneration, sinus elevation techniques, block grafts, alveolar distraction, and even the use of zygomatic implants. Although these techniques achieve good results and predictable prognoses, they remain complex surgical procedures that require multiple surgeries, prolong treatment, involve considerable post-operative discomfort for the patient, longer recovery periods, and increased economic cost—and even then, there is no guarantee of a successful outcome due to the higher rate of complications than with conventional implant placement in other areas [7,8,9,10,11,12,13,14]. To overcome this unfavorable situation, alternatives to implants of conventional length have been introduced: short implants, which do not require advanced surgical techniques for placement; they also reduce treatment time, minimize discomfort, and obtain successful functional outcomes, reduce surgical morbidity, and do not interfere with the anatomical structures adjoining posterior maxillary bone [13].

Short implants have been shown to suffer a significantly lower rate of complications and a success rate similar to implants of conventional length placed without applying additional (complex) techniques [7]. Numerous authors have concluded that the use of short implants offers an effective solution for patients with severe maxillary atrophy, and achieves high long-term survival rates of between 92% and 99% [7,10,11,12,13,14,15,16,17,18,19,20,21,22,23,24,25,26,27,28,29,30,31,32].

Although numerous articles have investigated the use of short implants, studies of splinted short implants are lacking [7,14]. Thus, the aim of this work was to analyze and compare the influence of extra-short dental implants (of 4 mm lengths), short dental implants (of 6 mm lengths), and conventional dental implants (of ≥8 mm lengths) supporting two splinted crowns (bridges) on marginal bone loss, as well as the influence of implant position (posterior or anterior position within the pair of splinted implants) on marginal bone loss, the differences between mesial and distal interproximal bone loss, and the survival rates of implants of different length. The work’s null hypotheses were that (H_0_) peri-implant bone loss with extra-short (4 mm) and short (6 mm) implants would be greater than implants of conventional length (>8 mm) and that there would be differences in marginal bone loss between extra-short (4 mm) and short (6 mm) implants; that (H_1_) implant positions (anterior/posterior) supporting prosthetic restorations (two splinted crowns) would not influence peri-implant bone loss; and that (H_2_) the survival rates of extra-short (4 mm) and short (6 mm) would be similar to that of implants of conventional length (≥8 mm).

## 2. Materials and Methods

### 2.1. Study Design

This retrospective clinical trial was conducted at the Faculty of Medicine and Dentistry, University of Valencia (Valencia, Spain) and fulfilled ethical guidelines established in the Declaration of Helsinki and the CONSORT Statement. The trial design was approved by the University of Valencia’s Ethics Committee for Experimental Human Research (Procedure No. H1508667813719). All patients were treated between June 2016 and December 2016, compiling data from medical records in June 2020. To allow the analysis of this retrospective clinical study, 3 groups with the same sample number and with similar implant placement dates were selected.

### 2.2. Inclusion Criteria

The inclusion criteria were: patients aged over 18 years and in good health and not presenting systemic pathologies; treated with pairs of Straumann^®^ Standard Plus (Institut Straumann AG, Basel, Switzerland) Regular Neck (RN) or Wide Neck (WN) tissue level Roxolid^®^ (Straumann) dental implants with the SLActive^®^ (Straumann) surface of 4, 6, and 8–12 mm lengths supporting two splinted crowns; they should be located in the area of the first and second maxillary molars; the antagonist was supposed to be the natural tooth; surgery had to be performed without the need for guided bone regeneration and the patient had to have post-loading periapical radiographs and others taken after 36 months. All patients gave their informed consent to take part in the study.

### 2.3. Clinical Procedure

This clinical trial included 24 patients treated consecutively with a total of 48 dental implants: eight RN 4.8 mm diameter 4 mm length extra-short dental implants (Ref.: 033.044S; Straumann); eight WN 4.8 mm diameter and 4 mm length extra-short dental implants (Ref.: 033.045S Straumann); ten RN 4.8 mm diameter and 6 mm length short dental implants (Ref.: 033.590S Straumann); six WN 4.8 mm diameter and 6 mm length short dental implants (Ref.: 033.610S Straumann); two RN 4.8 mm diameter and 8 mm length conventional dental implants (Ref.: 033.591S Straumann); eight RN 4.8 mm diameter and 10 mm length conventional dental implants (Ref.: 033.592S Straumann); two WN 4.mm diameter and 10 mm length conventional dental implants (Ref.: 033.612S Straumann); and four RN 4.8 mm diameter and 12 mm length conventional dental implants (Ref.: 033.583S Straumann). Patients were divided into groups according to the length of the implant they received: Group 1, eight patients restored with two 4 mm length extra-short dental implants supporting two splinted crowns (*n* = 16); Group 2, eight patients restored with two 6mm length short dental implants supporting two splinted crowns (*n* = 16); and Group 3, eight patients restored with two 8–12 mm length conventional dental implants supporting two splinted crowns (*n* = 16). In this way, each patient was rehabilitated with two dental implants of the same length and two implant-supported splinted crowns. All patients were monitored during a follow-up period of at least 36 months. All surgeries were performed under local anesthesia (4% articaine with 1:100,000 adrenalin (Inibsa^®^, Lliça de Vall, Barcelona, Spain) by the same clinician. The implants were placed in posterior maxillary ungrafted bone preparing the bed by means of the surgical drills, and drilling protocol recommended by the implant manufacturer. Surgical incision in the mucosa was supracrestal and intrasulcular in the area of the upper second molar by means of a full thickness flap without vertical release. All patients were treated in a single surgical session placing healing abutments on the implants’ prosthetic platforms. After implant placement and suturing, each patient received 500 mg of amoxicillin (Clamoxyl^®^, GlaxoSmithKline, Madrid, Spain) three times daily for 7 days, 600 mg of ibuprofen (Bexistar^®^, Laboratorio Bacino, Barcelona, Spain) to be taken as needed, and a 0.12% chlorhexidine mouthwash (GUM^®^, John O. Butler/Sunstar, Chicago, IL, USA) for use twice daily for two weeks. Gentle brushing with chlorhexidine toothpaste was also recommended. Sutures were removed 8–10 days after surgery. Prosthetic loading in the maxilla was carried out 10–12 weeks after implant placement (Figure 1).

The same lab technician fabricated all the restorations. These splinted crowns, corresponding to the maxillary first and second molars, were screwed onto the implants’ prosthetic platforms. All the crowns were made of chromium-cobalt with a feldspathic ceramic veneering (IPS d.SIGN, Ivoclar Vivadent, Schaan, Liechtenstein), milled using Computer-aided design/computer-aided manufacturing (CAD-CAM) technology. All screws were tightened applying 30 Ncm torque in accordance with the manufacturer’s specifications. The access holes of the screw-retained crowns were closed with Teflon pellets and a hybrid resin composite (Tetric-Ceram, Ivoclar Vivadent, Schaan, Liechtenstein).

### 2.4. Measurement Procedure

Marginal bone loss around implants was analyzed radiographically from digital periapical radiographs (Vistascan^®^, Dürr Dental, Bietigheim-Bissingen, Germany) taken with a parallelizer (XPC, Rinn, Dentsply Sirona^®^ Pt Ltd., St Leonards, NSW, Australia) for correct alignment. The radiographs were taken immediately after loading and at the end of the 36-month follow-up. Radiographic examination to analyze marginal bone changes was performed by a blinded examiner with the help of three-dimensional (3D) CAD software (Rhinoceros^®^ Robert Mcneel & Associates, Seattle, WA, USA). Marginal bone loss was measured on the mesial and distal interproximal surfaces of each pair of dental implants. The distance from the dental implant prosthetic platform to the most coronal part of the marginal bone crest was measured at both evaluation times (post-loading and 36 months later). Changes in marginal bone level between the time immediately after loading and the end of the 36-month follow-up were calculated and recorded. In addition, implant success rates were evaluated, applying Albrektsson’s success criteria to compare the study groups [33].

The protocol for measuring peri-implant bone loss was as follows:

Two reference points were located (clearly visible in radiographs and easily reproducible) at the join between the implant and the prosthetic crown, i.e., the line representing the prosthetic platform.Straumann^®^ Standard Plus (RN/WN) implants have a 1.8 mm polished collar. To scale the implant’s active portion, a perpendicular line was traced to measure the known length of the implant and the height of the polished collar.Having calibrated the radiographic image, differentiating between the polished collar and the real implant length, mesial, and distal reference points were established denominated as height 0.Lastly, to determine marginal bone loss on the mesial and distal aspects, a line was traced from height 0 to the most coronal bone. A value of 0 was given whenever the bone crest was higher than the reference point (Figure 2 and Figure 3).

### 2.5. Statistical Analysis

Statistical analysis of all variables was performed with SAS 9.4 statistical software (SAS Institute Inc., Cary, NC, USA). Descriptive statistics were expressed as means and standard deviation (SD) for quantitative variables. As the variables did not present normal distribution, marginal bone implant loss (mm) through dental implant positions and interproximal surfaces were compared between implant groups using the Kruskal–Wallis test. Non-parametric Brunner–Langer models were used to study interactions between groups, dental implant positions (anterior/posterior) and interproximal dental surfaces (mesial/distal). Statistical significance was set at *p* ˂ 0.05.

## 3. Results

Twenty-four patients (19 woman and 5 men) were enrolled in this clinical trial. The mean follow-up period was of 36.4 ± 1.6 months (minimum (min.) 34.0 months, maximum (max.) 38.3 months). None of the dental implants failed during the follow-up period; therefore, a 100% survival rate was achieved in all study groups.

Table 1 shows means and SDs for marginal bone implant loss (mm) on the mesial and distal interproximal surfaces of anterior and posterior dental implants of different dental implant lengths.

Overall median of bone loss in the sample was 0.41 (IQR 0.25–0.59). Bone loss in relation to implant length was as follows: extra-short implants (Group 1) obtained a median of marginal bone loss of 0.43 mm (0.24–0.55); short implants (Group 2) obtained 0.53 mm (0.40–0.62) and conventional length implants (Group 3) obtained 0.28 mm (0.21–0.49).

The 95% confidence intervals for the global mean bone loss medians of the 3 groups were calculated obtaining (0.00–0.73), (0.00–0.67), and (0.00–1.14) of extra-short, short and conventional length implants, respectively.

The Kruskal–Wallis test found no statistically significant differences in marginal bone loss between the three study groups (*p* = 0.421) (Figure 4 and Figure 5).

Analyzing bone loss in relation to interproximal area (mesial/distal), medians of marginal bone losses of 0.33 mm (0.00–0.42), 0.42 mm (0.11–0.69), and 0.14 mm (0.00–0.51) were observed on the mesial interproximal surfaces of extra-short, short, and conventional dental implants, respectively; medians of marginal bone implant losses of 0.46 mm (0.07–0.77), 0.50 mm (0.12–0.77), and 0.06 mm (0.00–0.68) were observed on the distal interproximal surfaces of extra-short, short, and conventional dental implants study groups, respectively.

Regarding marginal bone loss, according to the group and interproximal area (Figure 6), the values obtained suggest that bone loss was slightly greater in Group 2 (0.50 mm compared with 0.35 mm and 0.38 mm).

When marginal bone loss was analyzed in relation to implant position (anterior/posterior position within pairs of implants), medians of marginal bone losses of 0.35 mm (0.24–0.40), 0.39 mm (0.19–0.58) and 0.46 mm (0.26–0.81) were observed for anterior implants of extra-short, short and conventional dental implants, respectively; medians of marginal bone losses of 0.55 mm (0.16–0.76), 0.60 mm (0.37–0.80), and 0.03 mm (0.00–0.39) were observed for posterior dental implants of extra-short, short and conventional length implants, respectively (Figure 7).

Analyzing marginal bone loss in relation to implant length (extra-short, short, and conventional), interproximal area (mesial/distal), and implant position (anterior/posterior), a clearly irregular pattern was observed. In anterior positions, smaller differences between groups were observed than in posterior positions, particularly as in posterior positions implants of conventional length tended not to suffer any marginal bone loss at all. When the distribution of bone losses was compared between groups, implant position, and interproximal area (Brunner–Langer model ATS test) differences were identified although these depended on implant position (*p* = 0.028). Bone loss around more anteriorly placed implants was similar in the three groups, while in posterior positions, bone loss was much lower in the groups of implants of conventional length in comparison with short and extra-short implants. As no interaction was identified with the interproximal area, it may be concluded that this pattern existed independently of whether measurements were taken on the implants’ mesial or distal aspect.

As differences between groups were found among implants located in posterior positions, the Kruskal–Wallis test was applied to determine whether, in each interproximal area, and in the average of both interproximal areas together, there were differences in bone loss between groups; statistically significant differences were found. Thus, in posterior regions, it would appear preferable to place implants of conventional length due to the better management of the marginal bone loss produced on the distal aspect (*p* = 0.030).

Likewise, as implants of conventional length placed in posterior regions suffered less bone loss than short and extra-short implants, the distribution of bone loss was compared between implant position and interproximal area within the group of implants of conventional length (Brunner–Langer model ATS test). It was found that implants in posterior positions suffered significantly less marginal bone loss than implants in anterior positions (*p* = 0.026) regardless of whether bone loss was measured on the distal or mesial aspect (*p* = 0.331) (Figure 8).

## 4. Discussion

Recent years have seen increases in the use of short implants, despite some controversy in the literature as to the precise length that defines a “short” implant. According to some authors, an implant of ≤11 mm in length is considered short. Elsewhere, other authors define implants of <10 mm or <8 mm as short and implants of ≤6.5 mm as extra-short [10,13,15,16]. Nevertheless, most authors concur that that for an implant to be described as short its length should be less than 10 mm, considering that 10 mm long implants may be considered conventional due to the fact that their use has become widespread and routine [10]. The current tendency is to consider implants of <7 as short and those of smaller lengths as extra-short [16]. In the present study, the implants denominated as short were of 6 mm in length, extra-short of 4 mm and conventional length ≥8 mm. All of this has made it difficult to compare different investigations since there has been controversy regarding the length of the implants and their designation.

The study set out to evaluate marginal bone loss around implants supporting pairs of splinted crowns in the maxilla. Some authors suggest that biological overloading can be compensated by splinting implants as this will improve the distribution of mastication forces. It has been shown that the occlusal load to which splinted implants are subject undergoes a reduction in the forces generated within the body of the implant, and so a reduction in the stress transferred from the implant to the bone [21,24]. Rávida et al. [10] obtained statistically significant differences in favor of splinting short implants, as they found 3.3 times more prosthetic complications and 15.2 times greater probability of screw loosening in short single implants than others that were splinted. Torassa et al. [7] also splinted implants in a study comparing 4 mm implants with a longer implant in order to address the controversy in the literature regarding extra-short implants for rehabilitating single teeth in the maxilla; these authors and others [7,21,22] agree that splinting implants reduces biomechanical forces, including those forces deriving from the crown-to-implant ratio. When treating a case of major alveolar crest resorption—fairly common in the posterior maxilla—with splinted implants, the crown-to-implant ratio may be increased to overcome the vertical bone loss. This is an important factor in favor of splinting when using short implants, as the ratio will be higher than it should be. Various studies [7,10] have investigated crown-to-implant ratio in short and extra-short implants compared with implants >8 mm, showing that this factor did not influence the stability of the bone crest or implant survival. However, studies that have set out to determine whether or not to splint implants are few, and some have not found differences between splinting implants and rehabilitating the edentulous space with single crowns [21].

The present work focused on implant length and its influence on marginal bone loss and all the implants had 4.8 mm diameter in order to avoid more variables to analyze; but implant diameter is another factor for consideration that also influences bone loss. According to Slotte et al. [24], most bone stress occurs around the first few millimeters of the implant in the bone and their study concluded that, for this reason, diameter was a more important factor than length. This concurs with Loyola-González et al. [19], who concluded that most masticatory force is concentrated in the implant’s cervical portion, and so most of the occlusal load is transmitted to the bone from the implant’s cervical portion, and less force from the apical portion; this finding is also supported in work by Anitua et al. [21].

To compare the present results with the existing literature, it should be noted that, due to this study’s design and the limited number of clinical trials with similar methods, direct comparisons with the results of previous studies are not possible.

The present study comparing marginal bone loss and implant success between three groups of implants of different length did not find statistically significant differences between the groups. Similar results were obtained by Malmstrom et al. [22], who evaluated the success rate of short implants (6–8 mm) compared with implants of conventional length (11 mm), and did not find significant differences between the implant groups. A retrospective study by Mendonça et al. [31] analyzed the survival rate and bone loss around short (≤10 mm), splinted (*n* = 219), and non-splinted (*n* = 234) implants over 3–16 years. The peri-implant bone loss observed was not significantly different between splinted and non-splinted implants, although according to the descriptive statistics calculated, the splinted implants obtained better results. Clelland et al. [32], in a work with a similar design to the present study, grouped implants of different lengths (extra-short (6 mm); short (8–9 mm); standard length (11 mm)), according to whether they were splinted or not. After 36 months, bone loss for splinted implants was 0.68 ± 0.82 mm and 0.44 ± 0.58 mm for non-splinted implants with significant difference. However, descriptive statistics for the 6 mm splinted implants obtained slightly greater bone loss, although the difference did not reach statistical significance. This is a similar finding to the present study, as median values suggested that bone loss was slightly greater with 6 mm implants (Group 2), although the difference was not significant. Tabrizi et al. [23] aimed to determine whether there are significant differences in bone loss when more than one short implant are splinted. The sample was divided into three groups: Group 1, two splinted implants; Group 2, three splinted implants; Group 3, four splinted implants. They concluded that the higher the number of short splinted implants, the less the peri-implant bone loss will be. However, Pieria et al. [30] conducted a 3-year retrospective study, comparing the radiographic success rates of short implants (6–8 mm) compared with standard length implants (≥11 mm) placed with sinus lift in atrophic maxillae; higher bone loss was observed around conventional length (3.47 ± 0.89 mm) than short implants (3.18 ± 0.79 mm). Belindayi et al. [12] adopted a similar study design to Pieri, observing that short implants obtained similar results to implants of standard length in terms of survival and stability. Marginal bone loss around short implants (0.51 ± 0.54) was significantly less than around standard length implants after three years loading. Taschieri et al. [26] compared the behavior of short implants (6.5–8.5 mm) placed in the posterior maxilla with conventional length implants (≥10 mm) placed in atrophic posterior maxillae with sinus lift. It was concluded that, although bone losses were similar in the two groups, implants of conventional length suffered a much higher rate of complications.

Following a similar method to the present study, Torassa et al. [7] splinted 8 mm implants (in anterior position) to extra-short 4 mm implants (in posterior position) in the maxillary molar region. Although at an initial evaluation, extra-short implants showed better bone behavior than 8 mm implants, at the end of the follow-up period, mean bone loss for the extra-short implants was 1.14 ± 0.95 mm, and for the 8 mm implants it was 0.87 ± 0.41 mm. When these findings are extrapolated to the present work, there are clear differences, depending on whether implants were placed in anterior or posterior position. For anterior implants, the results were similar in the three groups, but in posterior implants, bone loss was slightly lower around implants of conventional length. In other words, analyzing the mesial and distal aspects of a posterior implant in each group as an overall evaluation, Group 3 showed slightly better results in terms of bone loss than the other groups, although without statistically significant difference. This could be due to the fact that the more posterior the area to be rehabilitated is, the poorer the bone quality will be, which will result in greater bone loss around the implant. So it would appear that in posterior areas, it may be preferable to place a longer implant, especially for better management of bone loss on the distal side. However, other authors, such as Anitua et al. [21], who set out to determine whether implant length in relation to position influences bone loss, did not find significant differences for 7 and 8.5 mm implants combined with implants of longer lengths (10–15 mm) placed in both maxilla and mandible, and monitored over a 10–12-year follow-up.

On the basis of the present results, the null hypothesis (H_0_) that peri-implant bone loss with extra-short (4 mm) and short (6 mm) implants would be greater than implants of conventional length (>8 mm), and that there would be differences in marginal bone loss between extra-short (4 mm) and short (6 mm) implants may be rejected, as no statistically significant differences were found between the groups. Moreover, marginal bone loss around short (6 mm) and extra-short (4 mm) implants was similar; although the median values obtained suggest that bone loss is greater in the group of short implants, the difference did not reach statistical significance. The second hypothesis (H_1_) that implant position (anterior/posterior) when pairs of implants support two splinted crowns would not influence peri-implant bone loss may also be rejected, as significant differences were found in Group 3 (conventional length implant), whereby the more posterior of the two implants suffered greater bone loss in comparison with Groups 1 and 2, in which no differences between posterior and anterior positions were detected. Lastly, as the implant survival rate for extra-short (4 mm), short (6 mm), and conventional length implants was 100%, the hypothesis that (H_2_) the survival rates of extra-short (4 mm) and short (6 mm) would be similar to that of implants of conventional length (≥8 mm) may be accepted.

To sum up, the use of extra-short (4 mm) and short (6 mm) implants supporting two splinted crowns in the atrophic posterior maxilla, obtained the same clinical outcomes as when using implants of conventional length (>8 mm). In spite of the study’s short follow-up and small sample size, the data produced provide clinical evidence in favor of the use of 4 and 6 mm implants. This approach may offer a viable alternative to more complex techniques for rehabilitating the atrophic posterior maxilla.

However, this study has some limitations. It was a retrospective clinical study, in future researches, it would be advisable to do a prospective randomized study. In addition, the number of the sample was 8 patients per group (analyzing 16 implants in each one), but due to the strict inclusion criteria requirements, it was difficult to get a larger sample that could include the last 3 years of clinical follow-up. It is important to emphasize that these implants were placed on remaining bone, at the posterior level, there is often the need to perform guided bone regeneration—another reason that also makes it difficult to meet the inclusion criteria. In future researches, the criteria may be looser to allow a greater number of samples analyzed.

## 5. Conclusions

Despite the limitations of the present study, it may be concluded that:Rehabilitation of the posterior maxilla with two splinted crowns on two extra-short (4 mm), short (6 mm) or conventional length (>8 mm) implants present similar marginal bone losses and survival rates after 3 years functional loading.Implants placed in posterior positions obtain better bone loss results than implants placed in anterior positions, regardless of the interproximal area where bone loss is measured.Implants of conventional length (≥8 mm) show less bone loss when placed in posterior positions than short (6 mm) and extra-short (4 mm) implants in posterior positions.Medium- and long-term longitudinal prospective studies are needed to support the use of short and extra-short implants in the maxillary posterior area.

## Figures and Tables

**Figure 1 ijerph-17-09278-f001:**
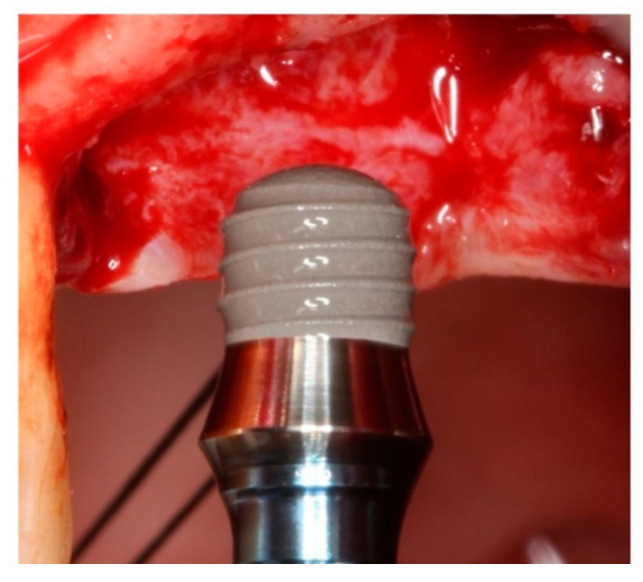
Surgical insertion of a 4 mm extra-short Straumann^®^ Standard Plus (RN) implant.

**Figure 2 ijerph-17-09278-f002:**
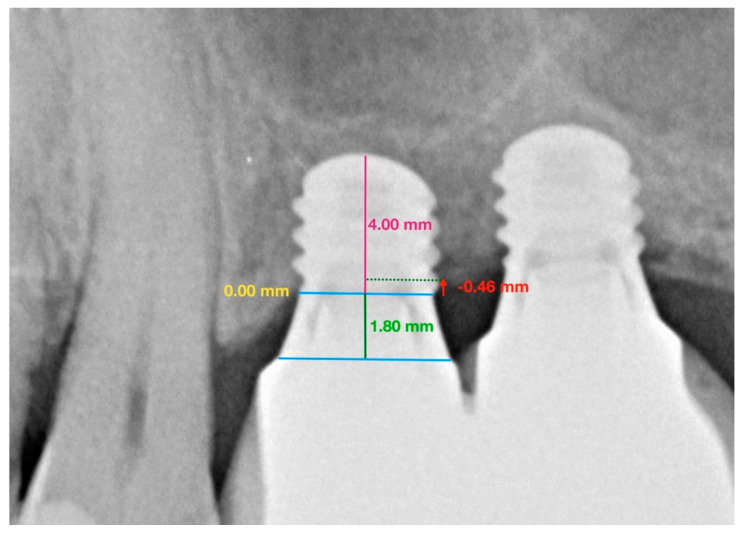
Group 1: Straumann^®^ Standard Plus (RN) extra-short 4 mm implants. Image shows the measurements taken from periapical radiographs to determine marginal bone loss around implants placed at positions 2.6 and 2.7 in the maxilla.

**Figure 3 ijerph-17-09278-f003:**
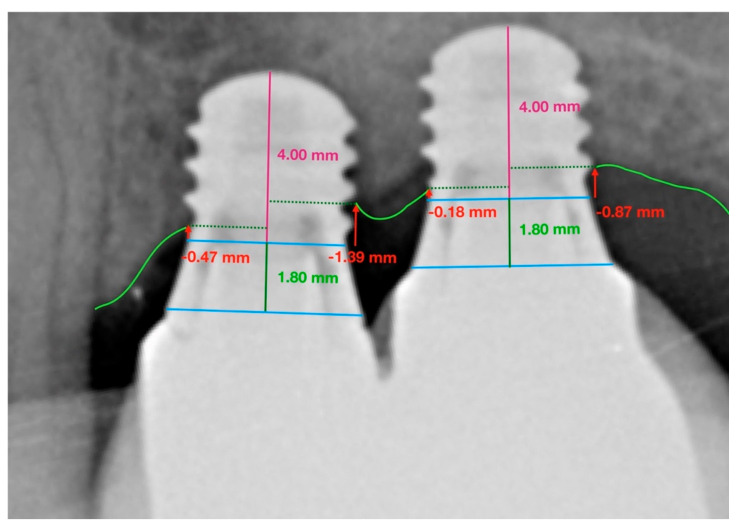
Successive measurements taken of Straumann^®^ Standard Plus (RN) extra-short 4 mm implants placed at positions 2.6 and 2.7 in the maxilla. Periapical radiograph taken at the end of the 36-month follow-up.

**Figure 4 ijerph-17-09278-f004:**
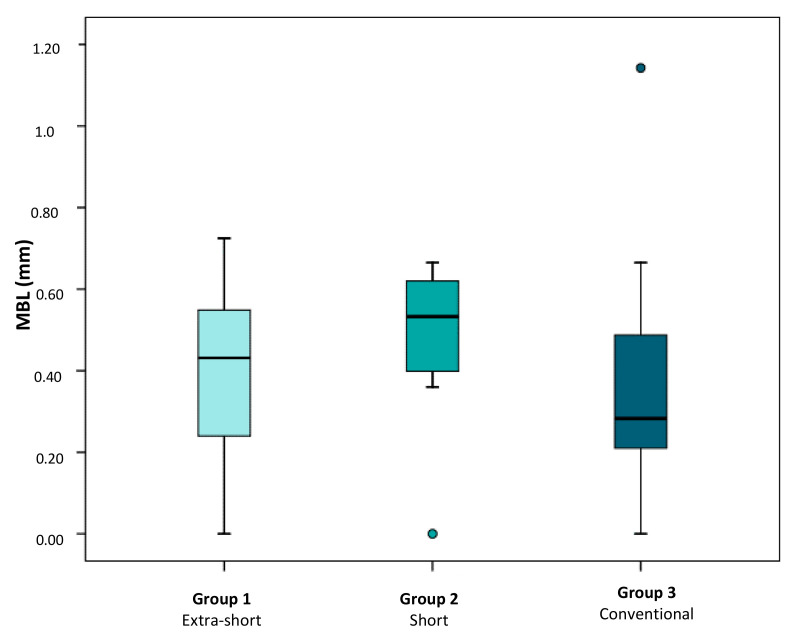
Box plot shows mean marginal bone loss obtained around extra-short, short, and conventional length implants. Atypical values are represented by circles.

**Figure 5 ijerph-17-09278-f005:**
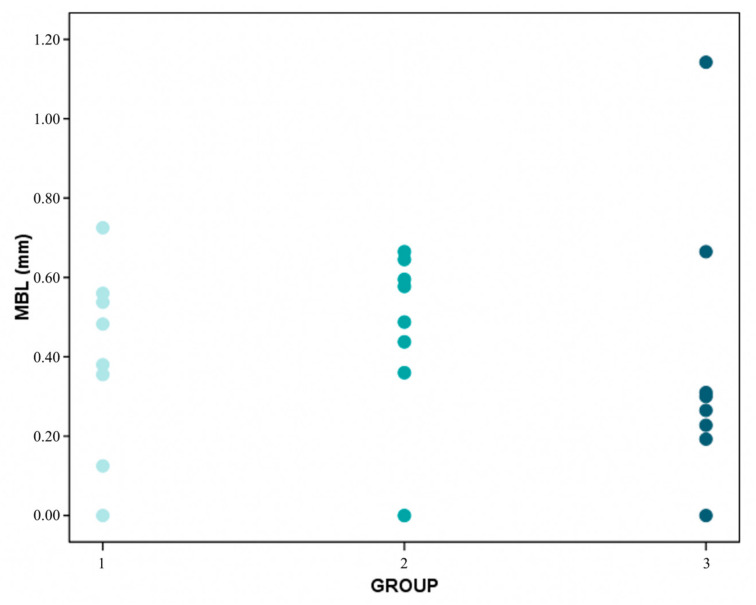
Distribution of MBL (mean bone level) average values per patient in groups 1 (extra-short), 2 (short) and 3 (conventional).

**Figure 6 ijerph-17-09278-f006:**
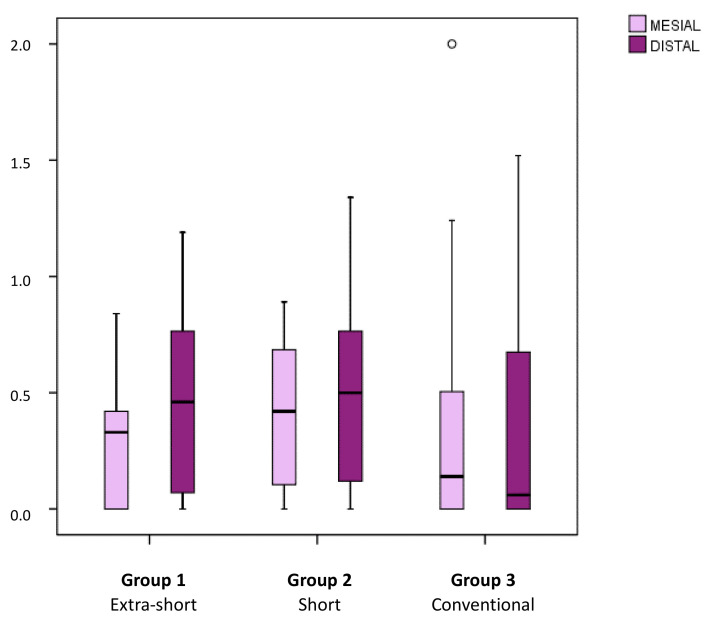
Box plots represent mean marginal bone loss of extra-short, short, and conventional length implants in relation to interproximal area (mesial/distal). Atypical values are represented by circles.

**Figure 7 ijerph-17-09278-f007:**
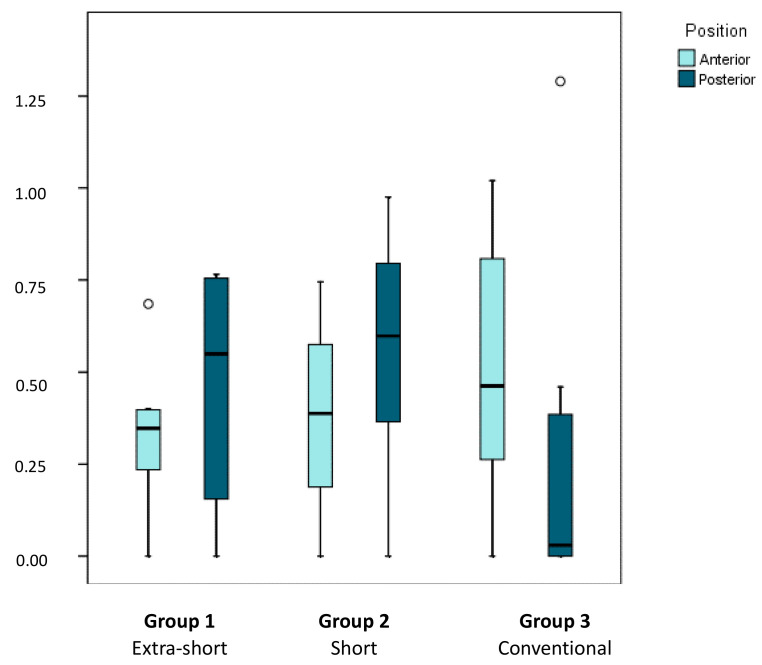
Box plot represents marginal bone loss in relation to implant position (anterior or posterior) for extra-short, short, and conventional dental implants. The horizontal line in each box represents the median value.

**Figure 8 ijerph-17-09278-f008:**
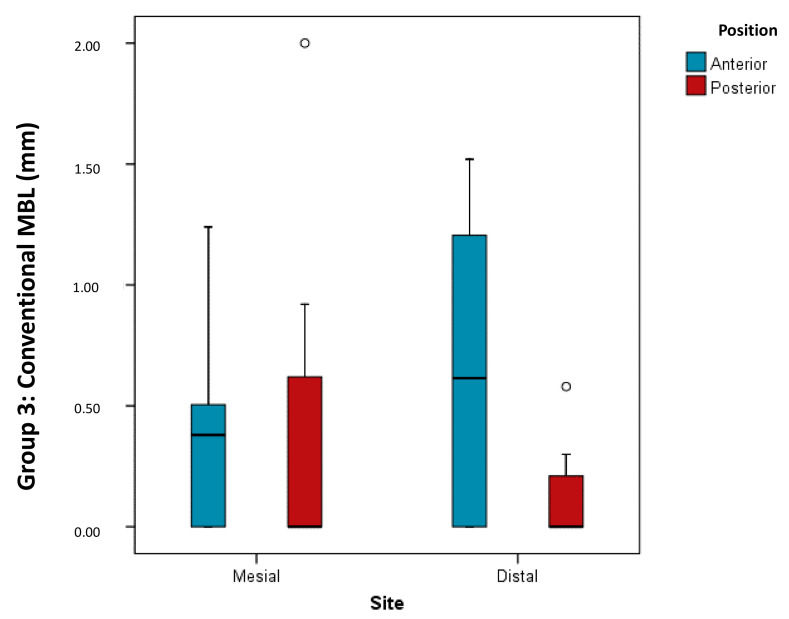
Box plot shows comparison of distribution of bone loss according to implant position and interproximal area analyzed within the group of conventional length implants.

**Table 1 ijerph-17-09278-t001:** Marginal bone implant loss (mm) on mesial and distal interproximal surface of anterior and posterior dental implants for extra-short (4 mm), short (6 mm), and conventional length (>8 mm) dental implants. Medians and Interquartile range (IQR).

		Anterior Dental Implant			Posterior Dental Implant		
		*n*	Median	IQR	*n*	Median	IQR
Extra-short implants (Group 1: 4 mm)	Mesial	8	0.32	0.12–0.40	8	0.35	0.00–0.77
	Distal	8	0.36	0.07–0.60	8	0.64	0.11–0.81
Short implants (Group 2: 6 mm)	Mesial	8	0.38	0.26–0.53	8	0.53	0.00–0.79
	Distal	8	0.37	0.00–0.65	8	0.55	0.41–1.12
Conventional implants (Group 3: ≥8 mm)	Mesial	8	0.38	0.00–0.51	8	0.00	0.00–0.62
	Distal	8	0.62	0.00–1.21	8	0.00	0.00–0.21

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
