# Peer review of "Comparative Analysis of Peri-Implant Bone Loss in Extra-Short, Short, and Conventional Implants. A 3-Year Retrospective Study"

_ijerph, 2020, doi:10.3390/ijerph17249278_

Round 1

Reviewer 1 Report

Dear Authors, I would suggest to review the title as you plan the research as a comparative study mainly on Peri implant bone loss, and your title refers to Prosthetic rehabilitation, you might check it, or consider to give more data on the prosthetic issue that is only refered as splinted crowns.

In Matherial and methods, you describe the implants diameters, please check and correct as there is no 6,5 mm diameter implant in Straumann, there is the WN which has the neck of 6,5 but the implant diameter is 4,8 mm, be careful there and review it.

Exclusion and inclusion criteria of the patients have to be better described.

You should diferentiate the different diameters in each group, as there might be a diference between 4,1 to 4,8 mm on clinical behaviour!!

This issue was adressed in the discussion and should be supported by your results if you diferentiate between different implant diameters.

I did not write on the manuscript as the corrections has to do with the items I wrote above.

Nice work to improve and publish!!

Author Response

Response to Reviewer 1

Dear Authors, I would suggest to review the title as you plan the research as a comparative study mainly on Peri implant bone loss, and your title refers to Prosthetic rehabilitation, you might check it, or consider to give data on the prosthetic issue that is only refered as splinted crowns.

Our response: Dear reviewer, thank you for your comment. The title has been changed as you advise:

“Comparative analysis of peri-implant bone loss in extra-short, short and conventional implants. A3-year retrospective study.”

In Matherial and methods, you describe the implants diameters, please check and correct as there is no 6,5 mm diameter implant in Straumann, there is the WN which has the neck of 6,5 but the implant diameter is 4,8 mm, be careful there and review it.

Our response: Dear reviewer, thank you for your comment. Sorry for the confusion, it was a mistake. As you explain, all the implants have 4,8mm diameter. It has been corrected as follows:

“This clinical trial included 24 patients treated consecutively with a total of 48 dental implants: eight RN 4.8 mm diameter 4 mm length extra-short dental implants (Ref.: 033.044S; Straumann); eight WN 4.8 mm diameter and 4 mm length extra-short dental implants (Ref.: 033.045S Straumann); ten RN 4.8 mm diameter and 6 mm length short dental implants (Ref.: 033.590S Straumann); six WN 4.8 mm diameter and 6 mm length short dental implants (Ref.: 033.610S Straumann); two RN 4.8 mm diameter and 8 mm length conventional dental implants (Ref.: 033.591S Straumann); eight RN 4.8 mm diameter and 10 mm length conventional dental implants (Ref.: 033.592S Straumann); two WN 4.mm diameter and 10 mm length conventional dental implants (Ref.: 033.612S Straumann); and four RN 4.8 mm diameter and 12 mm length conventional dental implants (Ref.: 033.583S Straumann).”

Exclusion and inclusion criteria of the patients have to be better described.

Our response: Dear reviewer, thank you for your comment. The inclusion criteria have been improved as you advise:

“The inclusion criteria were: patients aged over 18 years and in good health and not presenting systemic pathologies; treated with pairs of Straumann® Standard Plus (Institut Straumann AG, Basel, Switzerland) Regular Neck (RN) or Wide Neck (WN) tissue level Roxolid® (Straumann) dental implants with the SLActive® (Straumann) surface of 4, 6 and 8-12mm lengths supporting two splinted crowns located in the area of the first and second maxillary molars; with post-loading periapical radiographs and others taken after 36 months. Patients younger than 18 years, patients presenting systemic pathologies, or who had been treated with other types of dental implant or restorations, or did not have follow-up periapical radiographs available were excluded. All patients gave their informed consent to take part in the study.”

You should diferentiate the different diameters in each group, as there might be a diference between 4,1 to 4,8 mm on clinical behaviour!!

Our response: Dear reviewer, thank you for your comment. It was a mistake; all the implants have the same diameter of 4.8mm. It has been corrected as follows:

“This clinical trial included 24 patients treated consecutively with a total of 48 dental implants: eight RN 4.8 mm diameter 4 mm length extra-short dental implants (Ref.: 033.044S; Straumann); eight WN 4.8 mm diameter and 4 mm length extra-short dental implants (Ref.: 033.045S Straumann); ten RN 4.8 mm diameter and 6 mm length short dental implants (Ref.: 033.590S Straumann); six WN 4.8 mm diameter and 6 mm length short dental implants (Ref.: 033.610S Straumann); two RN 4.8 mm diameter and 8 mm length conventional dental implants (Ref.: 033.591S Straumann); eight RN 4.8 mm diameter and 10 mm length conventional dental implants (Ref.: 033.592S Straumann); two WN 4.mm diameter and 10 mm length conventional dental implants (Ref.: 033.612S Straumann); and four RN 4.8 mm diameter and 12 mm length conventional dental implants (Ref.: 033.583S Straumann).”

This issue was adressed in the discussion and should be supported by your results if you diferentiate between different implant diameters.

Our response: Dear reviewer, thank you for your comment. There is no difference in diameters (as all the implants have 4.8mm diameter) that is why the discussion has not been modified. However, the explanation of the diameter has been improved to avoid confusion.

It has been added to avoid confusion as follows:

“The present work focused on implant length and its influence on marginal bone loss and all the implants had 4.8mm diameter in order to avoid more variables to analyze; but implant diameter is another factor for consideration that also influences bone loss.”

I did not write on the manuscript as the corrections has to do with the items I wrote above.

Our response: Dear reviewer, thank you very much. The corrections you have suggested have been made.

Nice work to improve and publish!!

Our response: Dear reviewer, thank you for your good comments about our work.

Reviewer 2 Report

To me, the title is a bit misleading.  The study is an observational retrospective study with 3 years of follow-up.  The study was not designed as a clinical study with 3 years of follow-up.  E.g., a randomized clinical trial with patients randomized to 3 different types of implants and followed up for 3 years.  Hence, I was a bit disappointed to learn the study was a convenience sample of patients who had 3 years of follow-up data.

Also, I must be misunderstanding the description of study design.  Based on the description of the study design, it seems that patients had to have follow-up periapical radiographs, and hence, by study design there would be no “lost to follow-up”.  I was confused why in the results it was stated that there was no lost to follow-up if this was an inclusion criterion for the study.

Also, I find it hard to believe that 24 consecutively treated patients would result in 3 groups of equal number of patients given different implant type.  

Justification that the sample size was sufficient to draw any meaningful conclusions with only 8 patients per group needs to be given.   Clearly, the sample sizes are very small and unlikely to be sufficient to draw any meaningful conclusions, beyond providing pilot data for performing in the future a well-designed study.  It is fine to report on a pilot study, but it should be described as a pilot study.

As stated in the Statistical Analysis by the authors, “the variable did not present a normal distribution”, and hence, it is unlikely the mean and standard deviation are appropriate summary measures of the data.  The median and interquartile range should be reported instead.

The Kruskal-Wallis test requires that all observations within group are independent.  Clearly this is not true, given each patient was contributing two implants.  Hence, all statistical results may be invalid.

Although it may have been appropriate to use the non-parametric Brunner-Langer  model to test for an interaction, and simple alternative is simply to compute the difference between the mesial and distal sites within a patient and use the Kruskal-Wallis test.  The non-parametric Brunner-Langer model is not even familiar to most statistician, and hence a reference for the method should be given if it decided the method needs to be used.

Hopefully, the Kruskal-Wallis test was not used to compare survival rates between the compare survival rates between groups and this is a merely a typo.  The Kruskal-Wallis test would not be an appropriate test to compare survival rates.  Given there were no failures and the sample sizes are incredibly small for comparing survival rates, it is fine to just report descriptive summaries (i.e., 100% survival) and not to perform statistical tests.

Given the very sample size, it possible to display all the data points in figures 4 to figures 7, and the actual data values should be added to all the boxplots.

Given the small sample sizes, the failure to find statistically significant differences is expected and the absence of significant differences may merely be a reflection of the low statistical power of the study to demonstrate a difference. Hence, study findings as presented do not necessarily support the authors conclusion that the data produced provide clinical evidence in favor of the use of 4 and 6 mm implants.  In order to make such claims in the present of negative findings, confidence intervals should be reported for the primary comparisons, so one can determine if the failure to show a difference between implant types is merely due to a lack of precision (i.e., low statistical power) or that the different implants performed similarly.

Author Response

Response to Reviewer 2

To me, the title is a bit misleading.  The study is an observational retrospective study with 3 years of follow-up.  The study was not designed as a clinical study with 3 years of follow-up.  E.g., a randomized clinical trial with patients randomized to 3 different types of implants and followed up for 3 years.  Hence, I was a bit disappointed to learn the study was a convenience sample of patients who had 3 years of follow-up data.

Our response: Dear reviewer, sorry for the inconvenience. The title has been changed as you advise in order to avoid confusion:

“Comparative analysis of peri-implant bone loss in extra-short, short and conventional implants. A3-year retrospective study.”

Also, I must be misunderstanding the description of study design.  Based on the description of the study design, it seems that patients had to have follow-up periapical radiographs, and hence, by study design there would be no “lost to follow-up”.  I was confused why in the results it was stated that there was no lost to follow-up if this was an inclusion criterion for the study.

Our response: Dear reviewer, thank you for your comment. Sorry for the confusion. As a retrospective study, the entire radiographic sequence had to be available in order to assess implants clinical behavior.  We wanted to say that since no implant failed, we didn't lose any patients either of the initial selected sample. But the text has been changed to avoid confusion:

None of the dental implants failed during the follow-up period; therefore, a 100% survival rate was achieved in all study groups. Moreover, no patient was lost to the follow-up.”

Also, I find it hard to believe that 24 consecutively treated patients would result in 3 groups of equal number of patients given different implant type.  

Our response: Dear reviewer, thank you for your comment. Sorry for the confusion, the study was a retrospective observational study. That is why it result in 3 groups of equal number. As it is a retrospective clinical study we intentionally chose the sample to compare the 3 groups with the same sample number.

It has been added as follows:

“To allow the analysis of this retrospective clinical study, 3 groups with the same sample number and with similar implant placement dates were selected.”

Justification that the sample size was sufficient to draw any meaningful conclusions with only 8 patients per group needs to be given.   Clearly, the sample sizes are very small and unlikely to be sufficient to draw any meaningful conclusions, beyond providing pilot data for performing in the future a well-designed study.  It is fine to report on a pilot study, but it should be described as a pilot study.

Our reponse: Dear reviewer, thank you for your comment. The analyzed sample consisted of 24 patients and 48 implants. The sample is not very large due to the inclusion criteria, which are so severe. It is very difficult to get more patients than those analyzed in this study. Only implants are tested to restore maxillary first and second molars and selected patients should not require guided bone regeneration. Also, the antagonist must be a natural tooth. That is why this study could be the basis for a future study with a much larger sample and longer follow-up time.

This is one of the limitations of the study and it appears in the text as follows:

“In addition, the number of the sample was 8 patients per group (Analyzing 16 implants in each one), but due to the strict inclusion criteria requirements, it was difficult to get a larger sample that could include the last 3 years of clinical follow-up. It is important to emphasize that these implants were placed on remaining bone, at the posterior level there is often the need to perform guided bone regeneration, that was another reason which also makes it difficult to meet the inclusion criteria.”

As stated in the Statistical Analysis by the authors, “the variable did not present a normal distribution”, and hence, it is unlikely the mean and standard deviation are appropriate summary measures of the data.  The median and interquartile range should be  reported instead.

Our response: Dear reviewer, thank you for your comment. Following your instructions, the mean and standard deviation have been replaced by median and interquartile range (IQR) in order to improve the statical analysis of the investigation.

It has been added following your suggestions as follows:

“Table 1. Marginal bone implant loss (mm) on mesial and distal interproximal surface of anterior and posterior dental implants for extra-short (4 mm), short (6 mm), and conventional length (>8 mm) dental implants. Medians and Interquartile range (IQR)

Anterior Dental Implant

Posterior Dental Implant

n

Median

IQR

n

Median

IQR

EXTRA-SHORT IMPLANTS (Group 1: 4 mm)

Mesial

8

0.32

0.12 – 0.40

8

0.35

0.00 – 0.77

Distal

8

0.36

0.07 – 0.60

8

0.64

0.11 – 0.81

SHORT IMPLANTS (Group 2: 6 mm)

Mesial

8

0.38

0.26 – 0.53

8

0.53

0.00 – 0.79

Distal

8

0.37

0.00 – 0.65

8

0.55

0.41 – 1.12

CONVENTIONAL IMPLANTS (Group 3: ≥8 mm)

Mesial

8

0.38

0.00 – 0.51

8

0.00

0.00 – 0.62

Distal

8

0.62

0.00 – 1.21

8

0.00

0.00 - 0.21

Overall median of bone loss in the sample was 0.41 (IQR 0.25-0.59). Bone loss in relation to implant length was as follows: extra-short implants (Group 1) obtained a median of marginal bone loss of 0.43 mm (0.24-0.55); short implants (Group 2) obtained 0.53 mm (0.40-0.62) and conventional length implants (Group 3) obtained 0.28 mm (0.21-0.49).”

The Kruskal-Wallis test requires that all observations within group are independent.  Clearly this is not true, given each patient was contributing two implants.  Hence, all statistical results may be invalid.Although it may have been appropriate to use the non-parametric Brunner-Langer  model to test for an interaction, and simple alternative is simply to compute the difference between the mesial and distal sites within a patient and use the Kruskal-Wallis test.  The non-parametric Brunner-Langer model is not even familiar to most statistician, and hence a reference for the method should be given if it decided the method needs to be used.

Our response: Dear reviewer, thank you for your comment. Indeed, the Brunner-Langer model was the methodology chosen because of the intra-subject correlation of the observations (two implants per patient). This model is specific for this type of situation, with a low sample size and versatility according to the type of scale of the variables (it is applicable to continuous non-normal, ordinal or binary variables). In essence, it is a nonparametric alternative to the conventional ANOVA model. We have added 3 articles where the mathematical structure of the model and its scope of application are explained in order to facilitate the understanding:

Erceg-Hurn DM, Mirosevich VM. Modern robust statistical methods: an easy way to maximize the accuracy and power of your research. Am Psychol. 2008 Oct;63(7):591-601. doi: 10.1037/0003-066X.63.7.591. PMID: 18855490.

Noguchi, Kimihiro & Gel, Yulia & Brunner, Edgar & Konietschke, Frank. (2012). nparLD: An R Software Package for the Nonparametric Analysis of Longitudinal Data in Factorial Experiments. Journal of Statistical Software. 50. 10.18637/jss.v050.i12.

Brunner, Edgar & Langer, Frank. (2000). Nonparametric Analysis of Ordered Categorical Data in Designs with Longitudinal Observations and Small Sample Sizes. Biometrical Journal. 42. 663 - 675. 10.1002/1521-4036(200010)42:6<663::AID-BIMJ663>3.0.CO;2-7.

Hopefully, the Kruskal-Wallis test was not used to compare survival rates between the compare survival rates between groups and this is a merely a typo.  The Kruskal-Wallis test would not be an appropriate test to compare survival rates.  Given there were no failures and the sample sizes are incredibly small for comparing survival rates, it is fine to just report descriptive summaries (i.e., 100% survival) and not to perform statistical tests.

Our response: Dear reviewer, thank you for your comment and sorry for the inconvenience. This was indeed a typographical error. No statistical test was applied to contrast whether the 3 survival rates (100%) were the same or not. It has been corrected to avoid confusions.

“As the variables did not present normal distribution, marginal bone implant loss (mm) through dental implant positions and interproximal surfaces and survival rates were compared between implant groups using the Kruskal-Wallis test.”

Given the very sample size, it possible to display all the data points in figures 4 to figures 7, and the actual data values should be added to all the boxplots.

Our response: Dear reviewer, thank you for your comment. Sorry for the inconvenience, but here it is a technical limitation. The software does not allow to put in the same chart boxes and individual values. It should be noted that in some of them (for example, Figure 4), each box shows the distribution of 8 cases per group; but in others (Figure 5), 16 cases are being represented per box (8 previous and 8 later). If any figure is not necessary, we could delete it. In case it is more to your liking, we have added another Figure. It has been added as follows:

Figure 5. Distribution of MBL average values per patient in groups 1 (extra-short), 2 (short) and 3 (conventional).”

Given the small sample sizes, the failure to find statistically significant differences is expected and the absence of significant differences may merely be a reflection of the low statistical power of the study to demonstrate a difference. Hence, study findings as presented do not necessarily support the authors conclusion that the data produced provide clinical evidence in favor of the use of 4 and 6 mm implants.  In order to make such claims in the present of negative findings, confidence intervals should be reported for the primary comparisons, so one can determine if the failure to show a difference between implant types is merely due to a lack of precision (i.e., low statistical power) or that the different implants performed similarly.

Our response: Dear reviewer, thank you for your comment. The 95% confidence intervals for the global mean bone loss medians of the 3 groups were calculated obtaining (0.00-0.73), (0.00-0.67) and (0.00-1.14) of extra-short, short and conventional length implants, respectively. Given the small sample size per group, these intervals coincide with the minimum-maximum range of each group. For this sample size (8 per group), the intervals for the median will always have the minimum and maximum limits as limits, so, following the suggestion of the reviewer, we have described the results with interquartile ranges (IQR).

It has been added following the reviewer’s suggestion as follows:

“The 95% confidence intervals for the global mean bone loss medians of the 3 groups were calculated obtaining (0.00-0.73), (0.00-0.67) and (0.00-1.14) of extra-short, short and conventional length implants, respectively.”

Reviewer 3 Report

This paper presents an experimental study to evaluate the influence of implant length on marginal bone loss comparing 16 implants of 4 mm, 6 mm and >8 mm supporting two splinted crowns.

The is of relative interest (out the scope?) for the readers of the journal, is written clearly, and is easy to follow. My main concern about this paper is that dozens studies like these have been published in the scientific literature, so I do not see what is novel here. However, the authors claim that “Although numerous articles have investigated the use of short implants, studies of splinted short implants are lacking.” I do not have any elements to dispute this sentence.

Besides this major concern, the paper is fine. I suggest the following though:

  • The Figures must be changed and made more consistent. Some figures do not have proper labeling on the vertical axes. Other figures are too big and their caption is on the next page.
  • Despite the enormous literature on the subject, there are few references cited.
  • In the Conclusions, the author wrote: “Despite the limitations of the present study”. What are the limitations of the study?
  • The Conclusion should be ended with 1-2 sentences about potential paths for future studies.

Author Response

Response to reviewer 3

This paper presents an experimental study to evaluate the influence of implant length on marginal bone loss comparing 16 implants of 4 mm, 6 mm and >8 mm supporting two splinted crowns. The is of relative interest (out the scope?) for the readers of the journal, is written clearly, and is easy to follow. My main concern about this paper is that dozens studies like these have been published in the scientific literature, so I do not see what is novel here. However, the authors claim that “Although numerous articles have investigated the use of short implants, studies of splinted short implants are lacking.” I do not have any elements to dispute this sentence.

Our response: Dear reviewer, thank you for your comment. The nomenclature in recent years has been modified with respect to what short implant refers to, since 4mm implants a few years ago were not offered by the industry to clinicians. It's possible that articles from a few years ago regarding what were called short implants, were not considered short implants nowadays. Furthermore, the literature is scarce, especially comparing extra-short, short and conventional implants. In the scientific literature there are dozens studies that analyze the behavior of short implants. It should be noted that we are not evaluating their clinical behavior or if an extra-short implant works well.  Our objective is to evaluate, analyze and compare the influence of extra-short dental implants of 4mm length, short dental implants of 6mm length and conventional dental implants of ≥ 8mm length supporting two splinted crowns (bridges) on marginal bone loss, as well as the influence of implant position (posterior or anterior position within the pair of splinted implants) on marginal bone loss, the differences between mesial and distal interproximal bone loss, and the survival rates of implants of different length.   This will allow to see if the extra-short implants work the same, better or worse than the short and conventional ones. As well as comparing them to each other.

It has been explained as follows:

“Recent years have seen increases in the use of short implants despite some controversy in the literature as to the precise length that defines a ‘short’ implant. According to some authors, an implant of ≤11 mm in length is considered short. Elsewhere, other authors define implants of <10 mm or <8 mm as short and implants of ≤6.5 mm as extra-short [10,13,15,16]. Nevertheless, most authors concur that that for an implant to be described as short its length should be less than 10 mm, considering that 10 mm long implants may be considered conventional due to the fact that their use has become widespread and routine [10]. The current tendency is to consider implants of <7 as short and those of smaller lengths as extra-short [16]. In the present study, the implants denominated as short were of 6 mm in length, extra-short of 4 mm and conventional length ≥8 mm. All of this has made it difficult to compare different investigations since there has been controversy regarding the length of the implants and their designation.”

Besides this major concern, the paper is fine.

Our response: Dear reviewer, thank you for your comment. Your suggestions have been taken into account.

 I suggest the following though:

  • The Figures must be changed and made more consistent. Some figures do not have proper labeling on the vertical axes. Other figures are too big and their caption is on the next page.

Our response: Dear reviewer, thank you for your comment. The figures have been improved as you advise.

  • Despite the enormous literature on the subject, there are few references cited.

Our response: Dear reviewer, thank you for your comment. The references have been revised as you advise.

  • In the Conclusions, the author wrote: “Despite the limitations of the present study”. What are the limitations of the study?

Our response: Dear reviewer, thank you for your comment. The limitations of the study have been added as follows:

“However, this study has some limitations. It was a retrospective clinical study, in future researches, it would be advisable to do a prospective randomized study. In addition, the number of the sample was 8 patients per group (Analyzing 16 implants in each one), but due to the strict inclusion criteria requirements, it was difficult to get a larger sample that could include the last 3 years of clinical follow-up. It is important to emphasize that these implants were placed on remaining bone, at the posterior level there is often the need to perform guided bone regeneration, that was another reason which also makes it difficult to meet the inclusion criteria. In future researches the criteria may be looser to allow a greater number of samples analyzed.”

  • The Conclusion should be ended with 1-2 sentences about potential paths for future studies.

Our response: Dear reviewer, thank you for your comment. Your suggestion has been taken in account and it has been added as follows:

“ Medium- and long-term longitudinal prospective studies are needed to support the use of short and extra-short implants in the maxillary posterior area”

Round 2

Reviewer 2 Report

The authors have addressed my concerns about the original manuscript.

Reviewer 3 Report

I still believe that the paper does not meet the quality and novelty necessary to be published in a peer-reviewed journal. And I still believe that the paper is out of scope of this journal. I still believe the Introduction needs major improvement to describe adequately the state-of-the-art and adding 9 references without proper elaboration is a gross attempt to satisfy the reviewer rather than really improve the quality of the manuscript.